# Molecular Docking and Site-Directed Mutagenesis of GH49 Family Dextranase for the Preparation of High-Degree Polymerization Isomaltooligosaccharide

**DOI:** 10.3390/biom13020300

**Published:** 2023-02-06

**Authors:** Huanyu Wang, Qianru Lin, Mingwang Liu, Wen Ding, Nanhai Weng, Hao Ni, Jing Lu, Mingsheng Lyu, Shujun Wang

**Affiliations:** 1Jiangsu Key Laboratory of Marine Bioresources and Environment/Jiangsu Key Laboratory of Marine Biotechnology, Jiangsu Ocean University, Lianyungang 222005, China; 2Co-Innovation Center of Jiangsu Marine Bio-Industry Technology, Jiangsu Ocean University, Lianyungang 222005, China

**Keywords:** molecular docking, site-directed mutagenesis, isomaltooligosaccharide, high degree of polymerization, GH49 family dextranase

## Abstract

The high-degree polymerization of isomaltooligosaccharide (IMO) not only effectively promotes the growth and reproduction of Bifidobacterium in the human body but also renders it resistant to rapid degradation by gastric acid and can stimulate insulin secretion. In this study, we chose the engineered strain expressed dextranase (PsDex1711) as the research model and used the AutoDock vina molecular docking technique to dock IMO4, IMO5, and IMO6 with it to obtain mutation sites, and then studied the potential effect of key amino acids in this enzyme on its hydrolysate composition and enzymatic properties by site-directed mutagenesis method. It was found that the yield of IMO4 increased significantly to 62.32% by the mutant enzyme H373A. Saturation mutation depicted that the yield of IMO4 increased to 69.81% by the mutant enzyme H373R, and its neighboring site S374R IMO4 yield was augmented to 64.31%. Analysis of the enzymatic properties of the mutant enzyme revealed that the optimum temperature of H373R decreased from 30 °C to 20 °C, and more than 70% of the enzyme activity was maintained under alkaline conditions. The double-site saturation mutation results showed that the mutant enzyme H373R/N445Y IMO4 yield increased to 68.57%. The results suggest that the 373 sites with basic non-polar amino acids, such as arginine and histidine, affect the catalytic properties of the enzyme. The findings provide an important theoretical basis for the future marketable production of IMO4 and analysis of the structure of dextranase.

## 1. Introduction

Isomaltooligosaccharides (IMOs), also known as linear oligosaccharides, is a general term used for a class of oligosaccharides consisting of 2–5 glucose moieties bound by α-1,6 glycosidic bonds [1]. IMO has attracted significant attention in recent years due to its role in promoting the proliferation of intestinal probiotics and resisting caries production [2]. The main components of IMOs include isomaltose, panose, isomaltotriose, isomaltotetrasaccharide, and isomaltopentaose [3]. IMOs are regarded as the earliest developed product, the fastest growing, and the most versatile functional oligosaccharide, which can not only promote the proliferation of probiotic bacteria such as *Bifidobacterium* intestinalis but also indirectly hinder the growth of harmful bacteria and parthenogenic bacteria. It has therefore been widely used in medical, functional food, and food additive industries and is known as a novel bioglycogen in the 21st century [4]. The high degree of polymerization (DP ≥ 4) IMO is more promising because it can stimulate insulin secretion, does not affect the level of blood glucose, and is not easily digested by the gastric juice, thus making it easier to reach the intestinal tract [5].

The traditional method of producing IMOs is to use starch as the starting raw material and produce it through high-temperature liquefaction and medium-temperature saccharification steps, which consist of several procedures, consume much time, and comprise complicated operations. The final product obtained contains oligosaccharides with α-1,4 glycosidic bonds in addition to α-1,6 glycosidic bonds, which has lower purity [6]. In contrast, dextranase (EC 3.2.1.11), a member of the glycoside hydrolases (GH, EC 3.2.1) family, can specifically hydrolyze the α-(1,6) glycosidic bond of dextran, and the production of IMOs containing only an α-1,6 glycosidic bond can serve as an important strategy to solve the purity problem [7]. Based on the sequence, structural characteristics, and catalytic mechanism of the protein, and in combination with the protein structure database, the endo-dextranases (EC 3.2.1.11) were classified into two distinct families, namely, GH66 and GH49 [8]. In this study, the recombinant enzyme PsDex1711 used belongs to GH49 family dextranases, which consists of an N-terminal structural domain composed of a β-sandwich structure and a C-terminal catalytic structural domain. A total of 192 amino acids (Gly 33–Thr 225) are contained in the N-terminal structural domain whereas 125 amino acids (Gly 495–Trp 620) are contained in the C-terminal structural domain. The catalytic domain (Gln402–Asp424) is located in this gap, where Q402, D404, E407, D423, and D424 serve as the key conserved catalytic amino acids of the enzyme [9]. Among them, D423 functions as a general acid, which can effectively oxidize the glycoside with the substrate + 1 subsite into a hydrogen bond. On the contrary, E407 is a general base that can activate a water molecule, which then nucleophilically attacks the anomer carbon on the α-1,6 glycosidic bond of the dextran to achieve the breakage of the glycosidic bond [10].

Molecular docking is a theoretical simulation method that has been extensively used to study the potential interaction between the molecules (e.g., ligand and receptor) and to predict their binding modes and affinity. It has emerged as an important technique in computer-aided drug research in recent years [11]. Autodock vina is another widely used molecular docking program, developed by Olson’s group, which primarily applies a semi-flexible docking method that can facilitate the conformational change of the small molecules and evaluate the docking results based on the binding free energy [12]. Since AutoDock version 3.0, the optimization of energy has been performed using the Lamarckian genetic algorithm (LGA), which combines a genetic algorithm with a local search method to rapidly search the potential energy surface with a genetic algorithm and a local search method to conclusively optimize the potential energy surface [13].

Site-directed mutation refers to the introduction of the desired changes, including base addition, deletion, point mutation, etc., into the target DNA fragment by employing methods such as polymerase chain reaction (PCR) [14]. Fixed-point mutation can be divided into single-point mutation and multi-point mutation, which is a kind of rational design among protein-directed evolution techniques and can serve as a useful tool for the analysis of genes. It can be mostly used to modify and optimize target genes, explore regulatory sites of the promoters, and rapidly and efficiently improve the various traits and characterization of target proteins expressed by DNA [15]. The potential application of site-directed mutagenesis is not only widely used in the field of genetic engineering technology but has also been applied in agriculture to breed insect- and disease-resistant seeds and in medicine to correct genetic diseases and treat cancer as well as other chronic diseases [16].

In this study, high DP IMO4, IMO5, IMO6, and PsDex1711 were molecularly docked at the catalytic cleft, and the docking sites with low binding energy (more negative delta G value) were subjected to the site-directed mutagenesis, saturation mutagenesis, and two-site saturation mutagenesis to study the possible composition and enzymatic properties of the hydrolysate of the mutant enzymes. The study of the composition and enzymatic properties of the mutant enzyme hydrolysate can provide a sound theoretical basis for the structural study of GH49 family dextranases and the preparation of high DP IMOs.

## 2. Materials and Methods

### 2.1. Materials and Culture Conditions

The dextranase (PsDex1711) gene was cloned from *Pseudarthrobacter* sp. RN22, (sequence accession number PRJNA837017), which was screened from the marine mud collected from Rizhao, Shandong, China, and was maintained in this laboratory. A fast site-directed mutagenesis kit, plasmid microextraction kit, competent *Escherichia coli* DH5α, and *E. coli* BL21 (DE3) cells were purchased from Tiangen Biochemical Technology Co., Ltd. (Beijing, China). Kanamycin sulfate was purchased from Jiangsu Yugong Life Science&Technology Co., Ltd. (Lianyungang, China). IPTG (isopropyl-β-D-thiogalactoside), Dextran T70, and Dextran T500 were purchased from Shanghai Aladdin Biochemical Technology Co., Ltd. (Shanghai, China). TureColor two-color pre-stained protein marker (10 kDa–250 kDa) was purchased from Sangon Biotech (Shanghai, China) Co., Ltd. (Shanghai, China). Ni-NTA Superflow Agarose was purchased from Thermo.

The *E. coli* transformant-harboring recombinant plasmid was cultured in Luria Broth medium (LB medium) (10 g/L tryptone, 5 g/L yeast powder, 10 g/L NaCl, and kanamycin). It was maintained at a final concentration of 50 µg/mL).

### 2.2. Molecular Docking and Mutation Site Identification

The amino acid sequence of PsDex1711 was submitted to the NCBI database, and the highest amino acid sequence similarity was found between the enzyme and the dextranase AoDex of *Arthrobacter oxydans* KQ11 by using BLAST online comparison. The homology modeling of dextranase PsDex1711 by using SWISS-MODEL online software with AoDex (PDB code: 6nzs.1.A) as the template. Procheck, Errat, and Verify 3D were used to detect model rationality. The results were presented using PyMOL software.

Conserved residues in the catalytic region of PsDex1711 (Gln402–Asp424) were also found to be located in the catalytic cleft as determined by the AoDex catalytic domain range. Molecular docking of isomaltotetrasaccharide (IMO4), isomaltopentaose (IMO5) and isomalthexasaccharide (IMO6) with PsDex1711 in the catalytic cleft was carried out by using AutoDock vina molecular docking software to simulate the corresponding cleavage sites. The structures of IMO4, IMO5, and IMO6 were optimized by hydrogenation, calculation and average assignment of charges, and detection of rotational bonds and rotation centers before docking. Optimization of protein structure by hydrogen addition, dehydration, and charge calculation. The docking results were viewed using PyMOL software to select the sites possessing lower binding energy for mutagenesis based on the docking results. The mode of interaction between amino acids and IMOS was analyzed by using LigPlus software and Protein-Ligand Interaction Profiler (PLIP) online software.

### 2.3. Site-Directed Mutagenesis and Expression Purification of Recombinant Plasmids

A rapid site-directed mutagenesis kit was used to perform the site-directed mutagenesis of the lower binding energy sites using the unmutated PsDex1711 plasmid as a template. All the selected amino acids were mutated to alanine (Ala). The primers were designed according to the principles of single- and multi-site primer design, as indicated in Appendix A. The PCR reaction was set up as follows (50 μL): DNA template 2 μL, forward mutagenesis primer (10 μM) 2 μL, reverse mutagenesis primer (10 μM) 2 μL, 5× FastAlteration Buffer 10 μL, Fast Alteration DNA Polymerase (1.0 U/μL) 1 μL, RNase-Free ddH2O 33 μL. The reaction system used was as follows: 95 °C 2 min, 94 °C 20 s, 55 °C 10 s, 68 °C 2.5 min, 18 cycles, 68 °C 5 min. After PCR, 1 μL of Dpn I restriction enzyme (20 U/μL) at 37 °C for 1.5 h to digest the plasmid template.

After the digestion, the recombinant plasmid was transferred into the competent *E. coli* DH5α cells, and the single colonies in the plate were extracted by using the plasmid extraction kit. Thereafter, the recombinant plasmid was transferred into the competent *E. coli* BL21 (DE3) cells using the same method. After the estimation of OD_600nm_ = 0.6–0.8, the mixture was inoculated with new LB medium containing kanamycin 50 μg/mL at 4% inoculum and fermented at 3 °C, 180 r/min for 4–5 h. After the determination of OD_600nm_ = 0.6–0.8, IPTG was added to the medium at a final concentration of 0.5 mM, and the expression was induced at 16 °C, 180 r/min for 24 h and then centrifuged at 8000 r/min. After the centrifugation, the supernatant was discarded, and the bacteria were washed twice with 0.01 M PBS buffer, sonicated for 15 min, and then centrifuged at 8000 r/min for 10 min to obtain the crude enzyme solution. Enzyme activity was measured using the dinitrosalicylic acid method (DNS) method [17], and the enzyme activity units were defined (U/mL): the amount of enzyme solution required to release 1 μmol of isomaltose per minute of hydrolysis of dextran.

The crude enzyme solution containing His-tag obtained above was then purified by affinity chromatography on a nickel column, and 5 mL of crude enzyme solution was added to the nickel column at low temperature for 1 h to obtain the effluent solution and thereafter sequentially eluted with 20 mM, 40 mM, 60 mM, 80 mM, 100 mM, 200 mM, and 300 mM of imidazole-Tris-HCl buffer (final concentration of Tris-HCl 50 mM, the final concentration of NaCl 300 mM and sterilized by filtration with 0.22 μm membrane). The protein concentration of each eluate was determined using the BCA (Bicinchonininc acid) protein concentration assay kit, and the protein was then successfully purified using 10% SDS-PAGE (polyacrylamidegel electrophoresis) vertical electrophoresis.

### 2.4. Analysis of the Composition and Content of Mutant Enzyme Hydrolysate

The mutant enzyme solution was mixed 1:1 with the substrate 3% dextran T70 (pH = 6.0), and the reaction was carried out at 30 °C (1 h, 2 h, 3 h), boiling water bath for 5 min, and centrifuged at 8000 r/min for 5 min. The supernatant was filtered by 0.22 μm membrane and stored at 4 °C.

High-performance ion chromatography (HPIC) analysis: glucose (G), isomaltose (IMO2), isomaltotriose (IMO3), isomaltotetrasaccharide (IMO4), isomaltopentaose (IMO5), isomalthexasaccharide (IMO6), and isomaltheptasaccharide (IMO7) were purchased from Glycarbo as standards. The column used was DionexCarbopacTMPA200 (3 × 150 mm), the mobile phases: A: H_2_O; B: 130 mM NaOH D: 65 mM NaOH and 1025 mM NaOAc, a flow rate of 0.3 mL/min, and the injection volume was 25 μL. The detector was an electrochemical detector. The composition and content of each standard and the sample were analyzed according to the reported peak areas.

### 2.5. Saturation Mutations and Site Validation

The amino acid at the H373 site, where the product content was effectively altered, was saturated with the mutations by using a fast site-directed mutagenesis kit, i.e., mutated to all amino acids except its own amino acid and alanine. The primers (Appendix A) and vector were designed in the same way as described in Section 2.3, and the recombinant plasmid was transformed into competent *E. coli* cells. The mutant enzyme was expressed and then purified by using the same method, the enzyme activity was determined, and the hydrolysate was also analyzed by HPIC.

The amino acids at sites P371, Y372, S374, and F375, where the product of the saturation mutation was changed, were then selected for site-directed mutagenesis, and all amino acids were mutated to arginine (Arg). The enzyme activity was thereafter determined after the mutation, and expression purification was carried out by using a method similar to designing primers (Appendix A). The hydrolysate was finally analyzed using HPIC, and the mechanism leading to the change was elucidated.

### 2.6. Double-Site Saturation Mutations

After verifying the site mutation, it was confirmed that mutation of the H373 site amino acid to arginine could indeed markedly improve IMO4 production, and it was also found that the mutant enzyme N445A docked with IMO5 and IMO6 molecules also resulted in relatively higher IMO4 production, so the unmutated PsDex1711 plasmid was used as a potential template to mutate the H373 site amino acid to arginine and the N445 site amino acid was saturated and mutated to all amino acids for the saturation mutation of both the sites. The primer design is shown in Appendix A. Thereafter, an analysis of the content of the product after the saturation mutation and the mode of action between amino acids was carried out.

### 2.7. Analysis of the Enzymatic Properties of Mutant Enzymes

The mutant enzymes with altered hydrolysate: N445A, H373R, and H373R/N445Y were selected for optimal catalytic temperature as well as temperature stability and optimal catalytic pH and pH stability analysis.

Optimum catalytic temperature and temperature stability: The enzyme activity was measured at different temperatures (0 °C, 10 °C, 20 °C, 30 °C, 40 °C, 50 °C, 60 °C) with the substrate 3% dextran T500 (pH = 6.0); the enzymes were maintained at (20 °C, 30 °C, 40 °C, 50 °C) for (20 min, 40 min, 60 min, 80 min, 100 min, 120 min) The residual enzyme activity was detected afterward. All experiments were performed in triplicates.

Optimum catalytic pH and pH stability: The enzyme buffer (pH 4.0–9.0, final concentration 50 mM) was mixed with the substrate, which was 3% dextran T500 (pH = 6.0), and the enzyme activity was measured at the optimum temperature. The enzyme solution was mixed with the different pH and final concentration 50 mM buffer, and the residual enzyme activity was measured at the optimum temperature and retention time of 1 h. (acetate-sodium acetate buffer (pH 4.0, 5.0, 5.5, 6.0), PBS buffer (pH 6.0, 7.0, 7.5), and Tris-HCl buffer (pH 7.5, 8.0, 9.0) All experiments were performed in triplicates.

### 2.8. Statistical Analysis

AutoDock vina molecular docking results were analyzed using the software LigPlus software and the online software Protein-Ligand Interaction Profiler (PLIP) and viewed using PyMOL software. All graphs were plotted using Origin 2018 software. The formula for calculating enzyme activity by DNS method is as follows:$$\mathrm{Dextranase}\;\mathrm{activity}\left(U/\mathrm{mL}\right)=\frac{\mathrm{Reducing}\;\mathrm{sugar}\;\mathrm{quality}\left(\mu g\right)\times \mathrm{Dilutionmulitiple}}{\mathrm{Glucose}\;\mathrm{molecular}\;\mathrm{weight}\left(g/\mathrm{mol}\right)\times \mathrm{responsetime}\left(\min\right)\times \mathrm{Enzymeliquid}\;\mathrm{product}\left(\mathrm{mL}\right)}$$

## 3. Results

### 3.1. Identification of Mutation Sites

The sequence of PsDex1711 was analyzed by SWISS-MODEL, and AoDex 6nzs.1.A was used as the template for homology modeling (Figure 1a). According to the Ramachandran diagram (Appendix A), 85.7% of φ and ψ dihedral angles of the protein were found to be in the optimum region by Procheck, 13.3% in other allowable areas, 0.4% in the maximum allowable area, and only 0.4% in the nonallowable area. Errat scored 84.97. Therefore, the distribution ratios and the scores indicated that the 3D model obtained from the homology modeling of this enzyme is reliable and can be used for molecular docking experiments.

IMO4, IMO5, and IMO6 were molecularly docked with PsDex1711 in the catalytic cleft by using AutoDock vina molecular docking software in a semi-flexible docking mode, and the docking sites have been depicted in Figure 1b–d and Table 1. Five distinct binding sites were obtained: K427, H373, N445, S431, and Q402, all of which could form hydrogen bonds with IMOS with lengths less than 5 Å, and the lowest being 2.90 Å. All five docking sites displayed low binding energies, the lowest being −8.7 kJ/mol. It was also observed that both H373 and K427 could form salt bridges with other amino acids in the catalytic domain. The analysis of the electrostatic potential diagram (Appendix A) of this protein shows that the electron-rich region within the catalytic region is electrophilically active, which can accept a large amount of H^+^ and enhance the catalytic activity. Alanine remains as an uncharged hydrophobic amino acid compared to other amino acids. The smaller spatial site block of alanine can exhibit relatively less influence on the protein structure and can increase the size of the substrate binding pocket to a certain extent, which, in turn, can facilitate the binding of the enzyme to the substrate [18]. Therefore, all the site amino acids were mutated to alanine to explore the potential effect of the catalytic properties of this enzyme.

### 3.2. Acquisition and Expression of the Mutant Plasmids

Using the unmutated PsDex1711 plasmid as a template, PCR was employed to amplify and digest the plasmid template and then transfer it into the competent *E. coli* DH5α cells. The single colonies were then picked, and the plasmid was extracted, followed by the transfer of the recombinant plasmid into *E. coli* BL21 (DE3). Thereafter, the positive single colonies were picked, expression purified, and detected using 10% SDS-PAGE vertical electrophoresis with a protein loading volume of 3 μg, and the results are shown in Figure 2. A single band with a molecular weight of approximately 70 kDa was obtained after 300 mM imidazole elution, which was consistent with one before the mutation, and all enzymes were successfully mutated.

### 3.3. Hydrolysate of Molecular Docking Mutant Enzymes

The HPIC results of the mutant enzymes have been shown in Table 2 and Figure 3a, and the content of various oligosaccharides was determined according to the peak area ratio. It was observed that except Q402A, all the mutant enzymes only produced IMO3 and IMO4 by hydrolysate, and IMO4 accounted for about 60%. Q402 site was located in the catalytic cleft in the structure of PsDex1711, which is a key amino acid in the catalytic conserved structural domain of PsDex1711. When replaced with alanine, the carboxyl group changed to H and lost its ability to provide or accept H^+^ from the broad acid base, i.e., it lost its catalytic activity, and therefore no hydrolysate was produced.

Among them, 62.32% and 62.49% of IMO4 in the hydrolysate of H373A and N445A, respectively, and the yield of IMO4 was significantly increased in comparison with that of PsDex1711. It was also found that the enzymatic activity of H373A disappeared, but a small amount of hydrolysate could still be detected after 120 min of hydrolysis. The results of re-docking H373A with IMO4 have been shown in Figure 3b: it was noted that the hydrogen bonding disappeared, the salt bridge disappeared, but the binding energy increased from −6.4 to −6.0 kJ/mol, and hydrophobic interaction was formed. At the same time, the analysis of the amino acid structure revealed that the histidine side chain possessed one more imidazole group, which acts as a nucleophilic or electrophilic group on the electron-deficient or negatively charged center of the substrate, thus forming an unstable covalent intermediate, which is easily converted into a transmutation state. At the same time, the imidazole group can serve as an effective broad acid-base functional group, which can act as both a proton donor and a proton acceptor, being positively charged under neutral conditions and buffered under physiological conditions [19]. After mutation to alanine, the imidazole group was changed to H, thus modulating its catalytic properties. Since kinetic productivity analysis is critical to characterize the catalytic performance and capacity of the enzyme [20], the results of productivity analysis of mutant enzyme H373A are shown in Appendix A, and it was found that the yield of hydrolysate of mutant enzyme H373A increased after mutation, and the yield of IMO4 was always higher than IMO3.

The H373 site was close to the catalytic domain of PsDex1711 (Q402−D424), so it could exhibit some effect on its enzymatic activity. According to the previous study, PsDex1711 has high stability under weak alkaline conditions, whereas the isoelectric point decreased from 7.59 to 5.97 after mutation to alanine, thus changing from weak alkaline to weak acid. However, this advantage disappeared, and the enzyme’s ability to bind to the substrate dextran was markedly reduced, thus rendering the hydrolysate content lower. The remaining mutant enzymes did not have large differences in enzyme activity and hydrolysate types as well as the ratios, so the H373 site amino acid was selected for the saturation mutation to further investigate the mechanism of differences in hydrolysate formation.

### 3.4. Analysis of Saturated Mutant Enzyme Hydrolysate and Site Validation

The HPIC results of the saturation mutation of the H373 site amino acid have been shown in Table 3 and Figure 4a: Most of the mutated enzyme hydrolysates proportion remained unaltered, with 57−59% of IMO4 and 40−42% of IMO3, whereas some of the enzymes were not enzymatically active after the mutation. A significantly higher yield of 69.81% IMO4 was found at the mutant enzyme H373R, with a 7.49% higher IMO4 yield compared to H373A. The results of re-docking the mutant enzyme H373R with IMO4 have been depicted in Figure 4b,c: it was found that the binding energy decreased significantly from −6.4 to −6.8 kJ/mol after docking, the hydrogen bond length was basically unchanged, and the salt bridge length decreased from 4.64 to 4.55 Å, thus increasing the affinity with the substrate.

A study of the structural differences between histidine and arginine revealed that the side chain of arginine is a guanidinium group, which can potentially replace the imidazole group of histidine. The guanidinium group is a basic, easily hydrolyzed chemical group that can be completely protonated in the general physiological environment and can remain positively charged over a wide pH range. This special property can render the guanidinium group flexibility to easily form the special interactions between the ligand and receptor as well as enzyme and substrate by hydrogen bonding or electrostatic interactions, thus resulting in increased enzyme activity [21]. It was observed that after replacement with arginine, the isoelectric point increased from 7.59 to 10.07, and the basicity continued to increase, thereby enhancing the enzyme activity substantially and, thus, the hydrolysate content. It was also found that histidine, a positively charged basic amino acid, saturation mutation to a negatively charged acidic amino acid as well as most polar neutral amino acids, exhibited no enzymatic activity, but arginine, which is also a basic amino acid, affected the enzyme catalysis due to its different side chain groups, thus resulting in alteration in the conformation of the enzyme molecule, which was also related to the higher stability of PsDex1711 under the basic conditions.

P371, Y372, S374, and F375 site amino acids around the H373 site were mutated to arginine, and the HPIC results have been shown in Table 4 and Figure 4a: P371R, Y372R, and F375R possessed no enzymatic activity, and S374R mutated enzyme IMO4 yield increased to 67.31%. Upon comparing the structure, the serine side chain was found to contain a hydroxyl group, a typical polar group, which can effectively form hydrogen bonds with water and might inhibit its enzymatic activity. After replacing it with the guanidine group, the enzyme activity was enhanced by the same principle. It was also found that the isoelectric point increased from 5.68 to 10.07, the anion increased, and the enzyme’s ability to bind to the substrate was substantially enhanced, leading to an increase in IMO4 production. The site validation proved that basic amino acids such as arginine could play an important role in the production of IMO4 and in the mode of substrate cleavage.

### 3.5. Analysis of the Hydrolysate of Two-Site Saturable Mutant Enzymes

The results of the two-site saturation mutation of H373 site amino acid to arginine and N445 site amino acid to all amino acids have been shown in Table 5 and Figure 5a, and all the mutant enzymes except H373R/N445Y and H373R/N445G were found to be inactive. The relative enzyme activities of both the mutant enzymes were significantly decreased, 13.93% and 10.22%, respectively, probably because the H373 site and N445 site were both located near the catalytic domain of PsDex1711 (Q402–D424), and the mutation affected the size of the substrate binding pocket, which had a negative impact on the enzyme activity. Moreover, the IMO4 yield of the H373R/N445Y mutant enzyme was increased to 68.57%, which was 6.25% higher compared to the mutant enzyme H373A and 6.08% higher compared to the mutant enzyme N445A. The results of remolecular docking of H373R/N445Y with IMO4 have been depicted in Figure 5b,c). It was found that the hydrogen bonds between the R373 site and IMO4 increased to four with lengths of 4.10 Å, 3.05 Å, 4.00 Å, and 3.77 Å. The salt bridge length decreased markedly from 4.55 Å to 3.51 Å, which also exhibited some influence on the structural stability of the enzyme.

The structural analysis of tyrosine also showed that compared with the amide bond of the same uncharged polar amino acid asparagine, tyrosine acts as an aromatic polar α-amino acid containing a phenolic hydroxyl group, which is a dissociable group but has a high dissociation constant (pKa = 10.1), and hence, it remains essentially uncharged under the neutral conditions [22]. A π-π stacking interaction was formed between the phenyl ring of the tyrosine phenolic hydroxyl group and the phenyl ring of IMO4 so that there was a weak interaction between the electron-rich and electron-deficient molecules, which can have some influence on the catalytic properties of the enzyme.

### 3.6. Analysis of Enzymatic Properties of the Mutant Enzymes

As shown in Figure 6, compared with the unmutated PsDex1711, the optimum temperature of H373R was effectively reduced to 20 °C, but the optimum temperature of other mutant enzymes was still 30 °C. Analysis of its structural changes revealed that after the mutation to arginine, the side chain changed to the guanidine group, which could potentially maintain a positive charge in a wide range, and at the same time, the space at the edge of the catalytic domain was markedly expanded, which reduced the electrostatic effect [23] so that it could display high enzyme activity at the low temperature. In addition, the stability of the H373R/N445 mutant enzyme remained high at 30 °C, with 93.93% of enzyme activity at 120 min, whereas 74.26% and 72.25% of enzyme activity remained, respectively, for H373R and N445A.

It was also observed that the optimum pH of H373R and N445A was changed to 7.0, and the optimum catalytic pH increased to a neutral range compared with PsDex1711, but both maintained high enzyme activity under alkaline conditions. H373R was also found to have high stability under alkaline conditions, maintaining 75.1% of enzyme activity at pH = 9.0, which was a significant increase in pH stability in comparison with other enzymes. pH change can lead to alterations in the charged states of the substrate and enzyme molecules, thus affecting the dissociation of the relevant groups on the active site of the enzyme molecule. Interestingly, the shift in the direction of alkalinity was observed, with pH rising to 10.07, and the anion increased, thus indicating that the isoelectric point can exert a significant influence on the catalytic properties of the enzyme. From stability analysis, the H373R salt bridge length decreased from 4.64 to 4.55 Å. Although non-covalent interactions such as hydrogen bonding or electrostatic interactions of salt bridges have been known to be relatively weak interactions [24], small stable interactions can add up and play an important role in the overall stability of conformational isomers and aid in maintaining their high stability under basic conditions that can potentially render them of more specific industrial value.

## 4. Discussion

Molecular docking is a theoretical simulation method commonly used to predict the binding mode and affinity based on different intermolecular interactions [25]. In this study, recombinant dextranase PsDex1711 was docked with IMO4, IMO5, and IMO6 within the catalytic cleft to predict the exact cleavage site after the reaction of the enzyme with the substrate dextran to obtain a high degree of polymerization IMO. The potential use of molecular docking techniques to obtain binding sites has opened new avenues in the field of biological sciences. For example, Arya Devi KP [16] et al. used molecular docking to evaluate the possible binding mode of taurine at the SIRT1 variable site and reported that taurine displayed the most stable conformation with residues Q345, S442, and V445 in the SIRT1 protein with a minimum binding energy of −17.572 kJ/mol. Moreover, Selva Bilge [26] et al. investigated the interaction mechanism of ibrutinib molecules with dsDNA using a molecular docking technique and found that the binding of ibrutinib molecules to dsDNA was a groove-binding interaction mode, and the dominant interaction force was that of hydrogen bonding.

Protein-directed evolution techniques can be divided into three distinct strategies, namely, irrational design, semi-rational design, and rational design [27]. It has been established that fixed-point mutation as a rational design through computerized virtual mutation screening can quickly and accurately predict the target mutants [28]. In this study, all the sites obtained by the molecular docking were mutated to alanine, and the effect of each specific amino acid on the cleavage ability of the enzyme toward the substrate was studied. Similarly, Yang Peizhou [14] et al. modified the key amino acid sites of aflatoxin b1-degrading enzyme (TV-AFB1D), in which the E436A/H554A mutant possessed the highest enzyme activity with 1.84-fold higher activity in comparison to wild-type TV-AFB1D at an optimal pH of 7. Ruosi Fang [29] et al. used Discovery Studio software to perform the site-directed mutagenesis of ornithine transcarbamylase (OTC) and found that the catalytic activities of Q143W and H140A mutants were increased up to 2−3 times compared to the wild-type enzyme. The optimum temperature was reduced to 35 °C and the optimum pH to 8.5.

Glycoside hydrolases are a class of enzymes that can effectively hydrolyze glycosidic bonds (glycosidic bonds). Glycosidases can catalyze glycosidic reactions once the oxygen atom of the water molecule attacks the heterocapital carbon on the recipient glucose [30]. In the present study, site-directed mutagenesis of PsDex1711 revealed that the basic amino acids, such as arginine, could significantly affect the potential of the enzyme to cleave the substrate dextran. Similarly, Patcharapa Klahan [31] et al. used a plasmid containing SmDexTM coding sequence as a template for carrying out the site-directed mutagenesis of amino acids around its −3 and −4 subsites. They found that the mutants T558H, W279A/T563N, and W279F/T563N affected the hydrolytic activity of the enzyme at the −3 and −4 subsites, thereby reflecting a decrease in the substrate affinity at the −4 subsite and T558H increased the proportion of IMO5 in the hydrolysate after reaction with 4 mg/mL dextran. Annick Pollet [32] et al. performed site-directed mutagenesis of GH11 family xylanase A from *Bacillus subtilis* and found that the substitution of Tyr88 with Ala at the +3 site was able to produce large amounts of xylose. Thus, it can be speculated that the presence of a highly conserved aromatic residue at the +3 subsite can play a crucial role in xylanase activity and specificity. Castillo JdlM [33] et al. introduced a single mutation in the gene encoding *CGTase*, and the amino acid residue at position 137 proved to be its key catalytic site. This mutant demonstrated increased maltose production and a high degree of polymerization, mainly from maltopentaose to maltheptose. Christian Sonnendecker [34] et al. subjected a CGT enzyme from *Bacillus* sp. G-825-6 to site-directed mutagenesis at two distinct positions, with Y183W producing mainly CD8 and CD8 and larger CD being the only cyclic oligosaccharides produced. In addition, Jarunee Kaulpiboon [35] et al. prepared IMOs from cassava starch by using the mutant amylase in combination with Aspergillus niger transglycosidase. The highest yield of IMOs was obtained when 30% (*w*/*v*) soluble cassava starch was incubated with 120 units of amylase and 6 units of transglucosidase with mutant enzyme Y101S at 40 °C for 1 h. The presence of α-1,6, α-1,4, glycosidic bonds with a DP ≤ 9 was found to be significantly greater than that of commercial IMOs. However, in contrast, dextranase exclusively hydrolyzed α-1,6 glycosidic bonds, thus yielding IMOs with higher purity.

## 5. Conclusions

In summary, we have molecularly docked the GH49 family dextranases PsDex1711 and IMO4, IMO5, and IMO6 at the catalytic cleft and constructed the mutant enzymes K427A, H373A, N445A, S431, and Q402A. The enzymatic activity of H373A was observed to be greatly reduced. After the saturation mutation of the H373 site, the percentage of IMO4 in the hydrolysate of H373R reached up to 69.81%. At the same time, the yield of mutant enzyme S374R IMO4 also increased markedly to 64.31%. The results of the double-site saturation mutation showed that the mutant enzyme H373R/N445Y IMO4 yield rose to 68.57%. Analysis of the enzymatic properties revealed that the optimum temperature of H373R decreased to 20 °C, the optimum pH increased to 7.0, and more than 70% of enzyme activity was maintained under alkaline conditions. It was demonstrated that the guanidine group in the 373 site and the arginine side chain could substantially influence the catalytic properties of the enzyme, thus providing an effective strategy and market value for the production of high DP IMO.

## Figures and Tables

**Figure 1 biomolecules-13-00300-f001:**
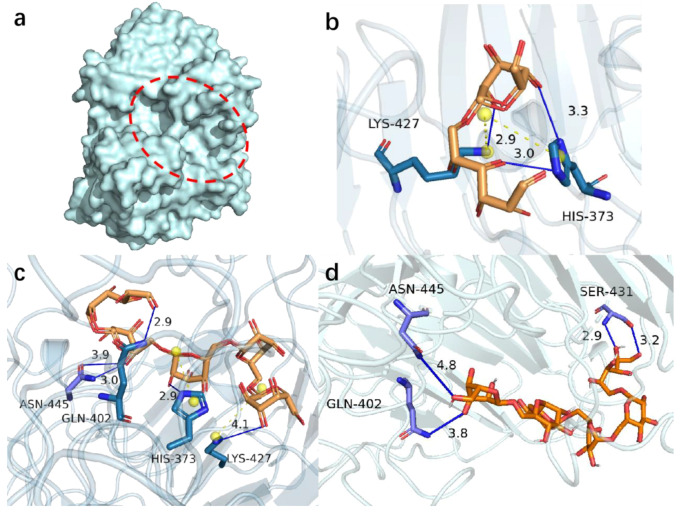
Molecular docking results of PsDex1711 (**a**) Location of catalytic cleavage in PsDex1711 structure shown; (**b**) IMO4 docking results; (**c**) IMO5 docking results; (**d**) IMO6 docking results. Hydrogen bonds have been depicted as blue lines, and salt bridges are shown as short yellow lines.

**Figure 2 biomolecules-13-00300-f002:**
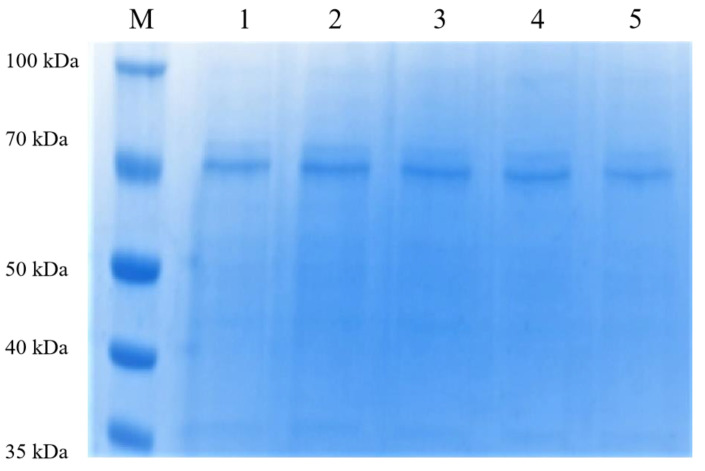
The 10% SDS-PAGE electrophoresis after purification of the mutant enzyme nickel column. 1–5: K427A, H373A, N445A, S431A, and Q402A 300 mM imidazole-eluting nickel column.

**Figure 3 biomolecules-13-00300-f003:**
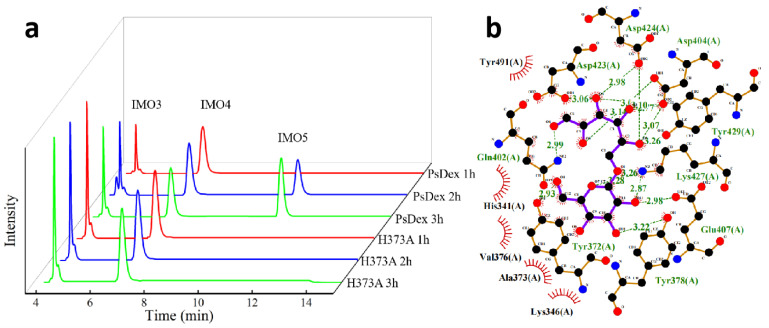
Analysis of H373A hydrolysate and molecular docking results. (**a**) HPIC analysis of H373A hydrolysate. (**b**) Results of the molecular docking between H373A and IMO4.

**Figure 4 biomolecules-13-00300-f004:**
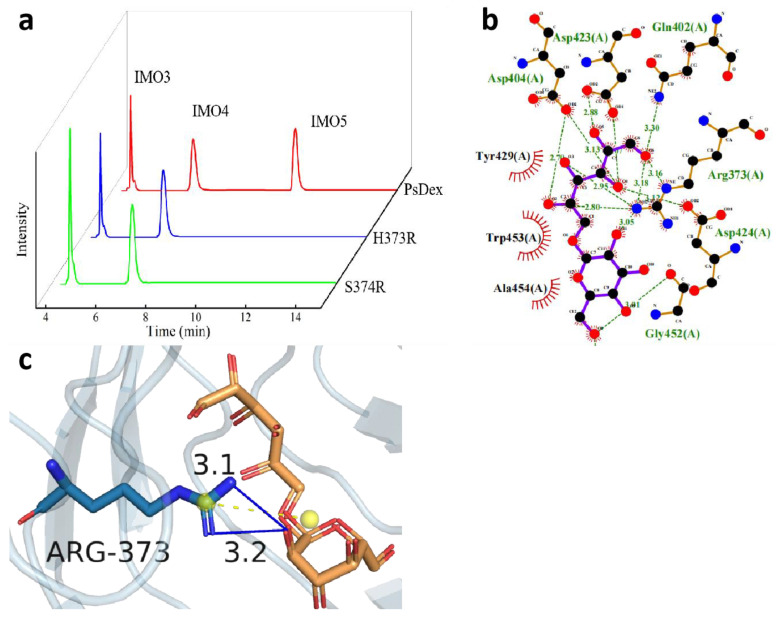
Analysis of the hydrolysate of H373R and S374R and the results of the molecular docking between H373R and IMO4. (**a**) HPIC analysis of mutant enzymes; (**b**) LigPlus molecular docking results are shown; (**c**) PyMOL depicting the mode of interaction of R373 with IMO4 in H373R. Hydrogen bonds have been depicted as blue lines, and salt bridges are shown as short yellow lines.

**Figure 5 biomolecules-13-00300-f005:**
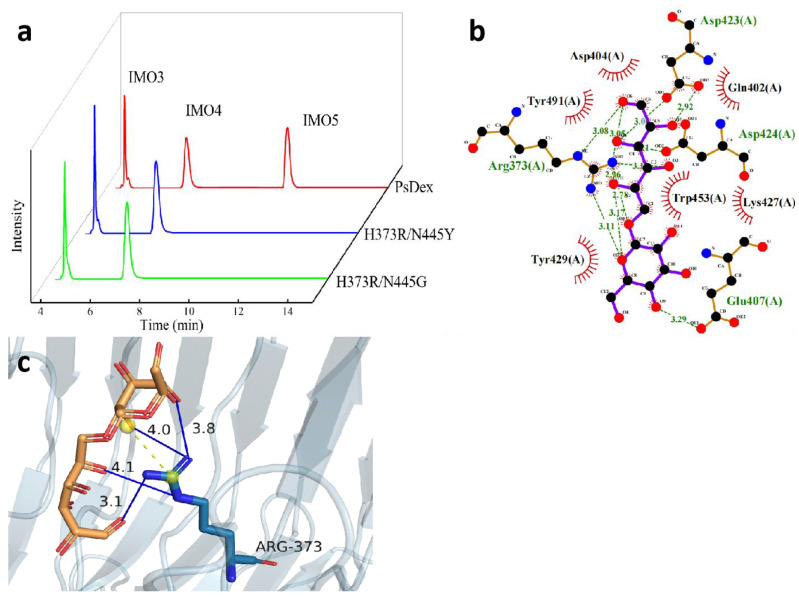
Analysis of the hydrolysate of the two-site saturated mutant enzymes and the results of molecular docking of H373R/N445Y with IMO4. (**a**) HPIC analysis of mutant enzymes; (**b**) LigPlus molecular docking results have been shown; (**c**) PyMOL depicting the mode of interaction of R373 with IMO4 in H373R/N445Y. Hydrogen bonds have been depicted as blue lines, and salt bridges are shown as short yellow lines.

**Figure 6 biomolecules-13-00300-f006:**
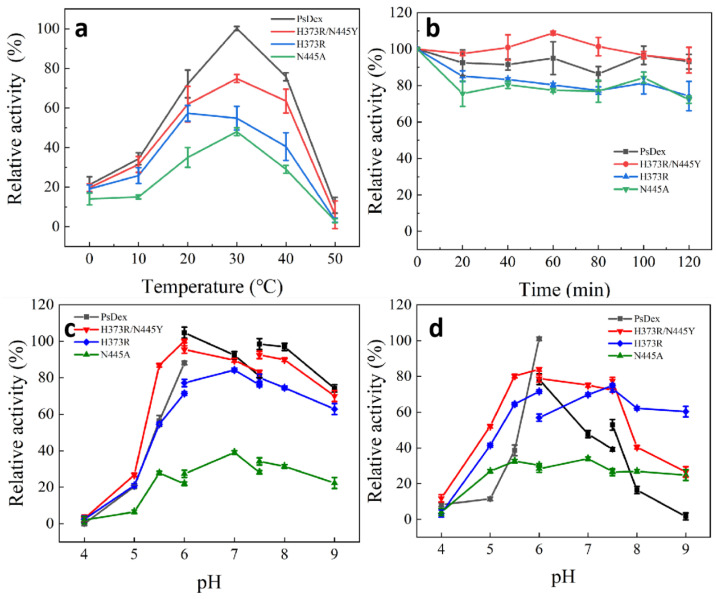
Analysis of enzymatic properties of the mutant enzymes. (**a**) Optimal catalytic temperature; (**b**) temperature stability; (**c**) optimal catalytic pH; (**d**) pH stability.

**Table 1 biomolecules-13-00300-t001:** PsDex1711 molecular docking results.

	Binding Energy (kJ/mol)	H-Bonds Length (Å) and Atomic Positions	Inhibition Constant (Ki) (mM)
**K427**	−6.4 (IMO4); −7.7 (IMO5)	NZ-O: 2.90; NZ-O: 4.10	8.73 (IMO4); 7.16 (IMO5)
**H373**	−6.4 (IMO4); −7.7 (IMO5)	NE2-O: 3.00; ND1-O: 3.30; NE2-O: 2.90	8.73 (IMO4); 7.16 (IMO5)
**N445**	−7.7 (IMO5); −8.7 (IMO6)	ND2-O: 3.00; OD1-O: 3.92; ND2-O: 4.80	7.16 (IMO5); 6.39 (IMO6)
**S431**	−8.7 (IMO6)	N-O: 2.90; OG-O: 3.20	6.39 (IMO6)
**Q402**	−7.7(IMO5); −8.7 (IMO6)	NE2-O: 2.94; NE2-O: 3.80	7.16 (IMO5); 6.39 (IMO6)

**Table 2 biomolecules-13-00300-t002:** Enzyme activity and ratio of hydrolysate of mutant enzyme.

	Relative Enzyme Activity (%)	Hydrolysate (%)
**PsDex1711**	100	IMO5: 49.18; IMO4: 24.50; IMO3: 26.32
**H373A**	30.20	IMO4: 62.32; IMO3: 37.68
**Q402A**	0	0
**K427A**	94.96	IMO4: 60.59; IMO3: 39.41
**N445A**	48.92	IMO4: 62.49; IMO3: 37.51
**S431A**	95.79	IMO4: 59.90; IMO3: 40.10

These data have been removed from the substrate residues, and the percentage has been calculated as the peak area of IMO.

**Table 3 biomolecules-13-00300-t003:** Saturated mutant enzyme activity and hydrolysate.

	Relative Enzyme Activity (%)	Hydrolysate (%)
**PsDex1711**	83.11	IMO5: 49.18; IMO4: 24.50; IMO3: 26.32
**H373A**	13.2	IMO4: 62.32; IMO3: 37.68
**H373R**	77.94	IMO4: 69.81; IMO3: 30.19
**H373F**	60.36	IMO4: 59.37; IMO3: 40.63
**H373C**	0	0
**H373G**	0	0
**H373Q**	0	0
**H373D**	0	0
**H373E**	0	0
**H373K**	80.04	IMO4: 58.92; IMO3: 41.08
**H373L**	97.27	IMO4: 57.52; IMO3: 42.48
**H373M**	92.58	IMO4: 57.30; IMO3: 42.60
**H373N**	97.09	IMO4: 58.85; IMO3: 41.15
**H373S**	100	IMO4: 57.82; IMO3: 42.18
**H373Y**	78.18	IMO4: 59.33; IMO3: 40.67
**H373T**	0	0
**H373I**	98.65	IMO4: 58.32; IMO3: 41.68
**H373W**	14.79	0
**H373P**	0	0
**H373V**	70.39	IMO4: 59.99; IMO3: 40.01

These data have been removed from the substrate residues, and the percentage has been calculated as the peak area of IMO.

**Table 4 biomolecules-13-00300-t004:** Site-validated mutant enzyme activity and hydrolysate.

	Relative Enzyme Activity (%)	Hydrolysate (%)
**PsDex1711**	87.25	IMO5: 49.18; IMO4: 24.50; IMO3: 26.32
**H373A**	17.31	IMO4: 62.32; IMO3: 37.68
**H373R**	81.94	IMO4: 69.81; IMO3: 30.19
**P371R**	0	0
**Y372R**	0	0
**S374R**	100	IMO4: 64.31; IMO3: 35.69
**F375R**	0	0

These data have been removed from the substrate residues, and the percentage has been calculated as the peak area of IMO.

**Table 5 biomolecules-13-00300-t005:** Enzymatic activity and hydrolysate of two-site saturated mutant enzymes.

	Relative Enzyme Activity (%)	Hydrolysate (%)
**PsDex1711**	100	IMO5: 49.18; IMO4: 24.50; IMO3: 26.32
**H373A**	30.20	IMO4: 62.32; IMO3: 37.68
**H373R**	91.42	IMO4: 69.81; IMO3: 30.19
**N445A**	48.92	IMO4: 62.49; IMO3: 37.51
**H373R/N445Y**	13.93	IMO4: 68.57; IMO3: 31.43
**H373R/N445G**	10.22	IMO4: 66.07; IMO3: 33.93

These data have been removed from the substrate residues, and the percentage has been calculated as the peak area of IMO.

## Data Availability

The datasets generated during and/or analyzed during the current study are available from the corresponding author upon reasonable request.

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
