# Peer review of "Molecular Docking and Site-Directed Mutagenesis of GH49 Family Dextranase for the Preparation of High-Degree Polymerization Isomaltooligosaccharide"

_biomolecules, 2023, doi:10.3390/biom13020300_

Round 1

Reviewer 1 Report

1.     The authors can incorporate the estimated inhibition constant (Ki) value in Table No. 1.

2.     In Table 1, why are there 2 categories of docking scores? Please categorize it.

3.     The authors are suggested to conduct molecular dynamic simulation (100ns) to determine inhibitor potential.

4.     Regulation of the figures (3b, 4b, 5b) could be better.

5.     The discussion may be improved by including more pertinent references.

Author Response

Dear Editor and Reviewers,

We are really appreciated for your kindly consideration to give us an opportunity to revise our manuscript entitled “Molecular Docking and Site-directed Mutagenesis of GH49 Family Dextranase for the Preparation of High Degree Polymerization Isomaltooligosaccharide” (biomolecules-2158057). We appreciate reviewers for the comments, and we have amended our manuscript according to the comments carefully. We have tried our best to revise our manuscript. Revised portions have been marked in yellow background in the manuscript. The main corrections in the paper and the responds to the reviewer’s comments are as following.

Responds to the reviewer 1’s comments:

1.     The authors can incorporate the estimated inhibition constant (Ki) value in Table No. 1.

Response: Thanks for your comment. We have added the inhibition constants in Table 1. 

2.     In Table 1, why are there 2 categories of docking scores? Please categorize it.

Response: Thanks for your comment. We have classified the docking scores according to the docking of different IMOs. They are in Table 1. Thanks. 

3.     The authors are suggested to conduct molecular dynamic simulation (100ns) to determine inhibitor potential. 

Response: Thank you very much for your comments. We have conduct molecular dynamic simulation (100 ns), and it will take over two months based on our computer equipment. We have analyzed the simulation result of 1ns. Also, we have tried to get the binding free energy, however, the system always reports an error. So, we analyzed the RMSD and RMSF results: the degree of variation of the protein conformation RMSD is very small, within 0.1 nm, and the degree of variation of RMSF is between 0.5-0.25 nm. Overall, the protein and the ligand can bind well and the overall conformation tends to be stable.

4.     Regulation of the figures (3b, 4b, 5b) could be better.

Response: Thanks for your comment. We have increased the resolution of Figures (3b, 4b, 5b) in the manuscript. Thanks. 

5.     The discussion may be improved by including more pertinent references.

Response: Thanks for your comment. We have increased the number of references in the discussion. They are in Lines 435,449 and 451.

We would like to express our great appreciation to you for comments on our manuscript again.  

Yours sincerely,

Huanyu Wang 

E-mail: wanghuanyuhk@163.com

Name: Jing Lu & Shujun Wang 

Reviewer 2 Report

The work described in this paper is interesting and of practical value. The authors have engineered rational mutations in an enzyme to increase the production of IMO4 while keeping maximum enzyme activity. It is a well-executed and well presented paper with the conclusions supported by the data.

I have minor comments below and a couple of major suggestions to further improve the manuscript.

From Topt data it seems the enzyme is cold-adapted. Therefore, it could be interesting if the activation thermodynamic parameters (deltaH# and delta S#) values for the native and mutants are given. This will dissect if  the decrease in activity is enthalpic or entropic provided authors already have Topt and Vmax/kcat data.

A productivity curve of the native and two top mutants will be very interesting as the formation of product (IMOs) as a function of time depends collectively on activity, stability, activation, and inhibition (product and substrate) and is the only valid biotechnological parameter (Int. J. Mol. Sci. 2022, 23(13), 6908; https://doi.org/10.3390/ijms23136908).

Line 17, abstract: the terminology throughout is confusing. engineered strain PsDex1711. A protein is not a strain. Only a bacteria can be a strain. Correct everywhere.

Line 20: yield of the mutant enzyme H373A IMO4 increased significantly. what does this sentence mean? Does the author mean the ability of H373A mutant to make IMO4? Using dextran? Why IMO4 is after the mutant? IMO4 is the product of the reaction. Correct everywhere.

Line 24: H373R was reduced to 20°C. Reduced from which temperature?

Abstract: Confusing. Should be rewritten.

Line 94: docking sites with low binding energy. The author means “high binding energy (more negative delta G values)?

Lines 233-234 & Table 1: It was also observed that both H373 and K427 site amino acids can form salt bridges with IMO4 and IMO5 structurally. This statement is scientifically not correct because a salt-bridge (ionic bond) is between two oppositely charged molecule centres like an NH3+ and COO-. There are no charged centres in IMO. These are H-bonds and not salt-bridges. Clarify.

Tables 3 & 4: why the activity of native enzyme is 83.11 %. It should be 100%? Clarify under all tables.

The use of word “mutase” is confusing because mutase is a completely different enzyme that catalyzes the movement of a functional group from one position to another within the same molecule. Clarify.

Author Response

Dear Editor and Reviewers,

We are really appreciated for your kindly consideration to give us an opportunity to revise our manuscript entitled “Molecular Docking and Site-directed Mutagenesis of GH49 Family Dextranase for the Preparation of High Degree Polymerization Isomaltooligosaccharide” (biomolecules-2158057). We appreciate reviewers for the comments, and we have amended our manuscript according to the comments carefully. We have tried our best to revise our manuscript. Revised portions have been marked in yellow background in the manuscript. The main corrections in the paper and the responds to the reviewer’s comments are as following.

Responds to the reviewer 2’s comments:

  1. From Topt data it seems the enzyme is cold-adapted. Therefore, it could be interesting if the activation thermodynamic parameters (deltaH# and delta S#) values for the native and mutants are given. This will dissect if the decrease in activity is enthalpic or entropic provided authors already have Topt and Vmax/kcat data.

Response: Thanks for your comment. We have conduct molecular dynamic simulation (100 ns), and it will take over two months based on our computer equipment. We have analyzed the simulation result of 1ns. Also, we have tried to get the binding free energy, however, the system always reports an error. So, we analyzed the RMSD and RMSF results: the degree of variation of the protein conformation RMSD is very small, within 0.1 nm, and the degree of variation of RMSF is between 0.5-0.25 nm. Overall, the protein and the ligand can bind well and the overall conformation tends to be stable.

  1. A productivity curve of the native and two top mutants will be very interesting as the formation of product (IMOs) as a function of time depends collectively on activity, stability, activation, and inhibition (product and substrate) and is the only valid biotechnological parameter (Int. J. Mol. Sci. 2022, 23(13), 6908; https://doi.org/10.3390/ijms23136908).

Response: Thanks for your comment. We investigated the productivity curves of mutant enzyme H373A and unmutated enzyme in Figure S3. The description of it is in 296-300 lines.

  1. Line 17, abstract: the terminology throughout is confusing. engineered strain PsDex1711. A protein is not a strain. Only a bacteria can be a strain. Correct everywhere.

Response: Thanks for your comment. We have changed it to dextranase PsDex1711. They are in Lines 17 and 102.

  1. Line 20: yield of the mutant enzyme H373A IMO4 increased significantly. what does this sentence mean? Does the author mean the ability of H373A mutant to make IMO4? Using dextran? Why IMO4 is after the mutant? IMO4 is the product of the reaction. Correct everywhere.

Response: Thanks for your comment. The statement is that the yield of IMO4 in the hydrolysate was increased by the mutant enzyme.  We have revised in the manuscript. Thanks.

Line 24: H373R was reduced to 20°C. Reduced from which temperature?

Response: Thanks for your comment. We have changed it to: the optimum temperature was reduced from 30°C to 20°C. They are in Lines 23-24. We have revised in the manuscript.

  1. Line 94: docking sites with low binding energy. The author means “high binding energy (more negative delta G values)?

Response: Thanks for your comment. We have added a note at the end of the sentence: more negative delta G values. They are in Lines 94-95.

  1. Lines 233-234 & Table 1: It was also observed that both H373 and K427 site amino acids can form salt bridges with IMO4 and IMO5 structurally. This statement is scientifically not correct because a salt-bridge (ionic bond) is between two oppositely charged molecule centres like an NH3+ and COO-. There are no charged centres in IMO. These are H-bonds and not salt-bridges. Clarify.

Response: Thanks for your comment. We have changed it to: the docking site amino acid residue forms a salt bridge interaction with an amino acid near the IMO. They are in Lines 243-245. This data was analyzed by the online software Protein-Ligand Interaction Profiler (PLIP).

  1. Tables 3 & 4: why the activity of native enzyme is 83.11 %. It should be 100%? Clarify under all tables.

Response: Thanks for your comment. We have set the highest enzyme activity value as 100% in each set of tables. In table 4, the S374R is 100%.

  1. The use of word “mutase” is confusing because mutase is a completely different enzyme that catalyzes the movement of a functional group from one position to another within the same molecule. Clarify.

Response: Thanks for your comment. We have revised it to mutant enzyme in the manuscript.

We would like to express our great appreciation to you for comments on our manuscript again. 

Yours sincerely,

Huanyu Wang

E-mail: wanghuanyuhk@163.com

Name: Jing Lu & Shujun Wang

Reviewer 3 Report

The manuscript biomolecules-2158057 reports in silico and experimental assays for some dextranases’ mutants. I recommend the publication of this work after minor revision as follows below:

1) Please, for the homology modeling provide the following data and, in some cases, the corresponding plots as supplementary material: GMQE, QMEAN Z-Score, Ramachandran plot, rotamer outliers (%), bad angles (%), RMSD, comparison with a non-redundant set of PDB structure, local quality estimate…….

2) Please, inform in section 2.2 of the structural optimization (energy-minimization) used to IMO4, IMO5, and IMO6 before molecular docking calculations.

3) For the in silico results, provide the electrostatic potential map of the protein and verify some trends based on this novel data.

4) Since dextranases reactions are required catalytic waters (10.1111/j.1432-1033.2004.04378.x, 10.1016/S0969-2126(03)00147-3), what is the reason for in silico calculations the authors did not consider it? Please, explain it in the main manuscript.

5) Please, explain in the manuscript how it is possible to have a salt bridge interaction between the detected amino acid residues with a non-charged ligand.

Author Response

Dear Editor and Reviewers,

We are really appreciated for your kindly consideration to give us an opportunity to revise our manuscript entitled “Molecular Docking and Site-directed Mutagenesis of GH49 Family Dextranase for the Preparation of High Degree Polymerization Isomaltooligosaccharide” (biomolecules-2158057). We appreciate reviewers for the comments, and we have amended our manuscript according to the comments carefully. We have tried our best to revise our manuscript. Revised portions have been marked in yellow background in the manuscript. The main corrections in the paper and the responds to the reviewer’s comments are as following.

Responds to the reviewer 3’s comments:

  1. Please, for the homology modeling provide the following data and, in some cases, the corresponding plots as supplementary material: GMQE, QMEAN Z-Score, Ramachandran plot, rotamer outliers (%), bad angles (%), RMSD, comparison with a non-redundant set of PDB structure, local quality estimate…….

Response: Thanks for your comment. We have added a description of the homology modeling, they are in Lines119-122 and 230-237. And added Ramachandran diagram in Figure S1.

  1. Please, inform in section 2.2 of the structural optimization (energy-minimization) used to IMO4, IMO5, and IMO6 before molecular docking calculations.

Response: Thanks for your comment. We have added instructions for optimizing the structure of IMOs, they are in Lines128-130.

  1. For the in silico results, provide the electrostatic potential map of the protein and verify some trends based on this novel data.

Response: Thanks for your comment. We have added protein electrostatic potential maps (Figure S2) and described them in lines 245-247.

  1. Since dextranases reactions are required catalytic waters (10.1111/j.1432-1033.2004.04378.x, 10.1016/S0969-2126(03)00147-3), what is the reason for in silico calculations the authors did not consider it? Please, explain it in the main manuscript.

Response: Thanks for your comment. We have optimized the 3D structure of the enzyme by dehydration and hydrogenation before docking to exclude the influence of other factors. They are in Lines130-131.

  1. Please, explain in the manuscript how it is possible to have a salt bridge interaction between the detected amino acid residues with a non-charged ligand.

Response: Thanks for your comment. We have revised in the manuscript: the docking site amino acid residue forms a salt bridge interaction with an amino acid near the IMO. They are in Lines 243-245. This data was analyzed by the online software Protein-Ligand Interaction Profiler (PLIP).

We would like to express our great appreciation to you for comments on our manuscript again. 

Yours sincerely,

Huanyu Wang

E-mail: wanghuanyuhk@163.com

Name: Jing Lu & Shujun Wang
